# Chip-scale sensor for spectroscopic metrology

Chunhui Yao [1,2,3], Wanlu Zhang[1,3], Peng Bao[1], Jie Ma[2], Wei Zhuo[2],
Minjia Chen [1], Zhitian Shi[1], Jingwen Zhou[2], Yuxiao Ye[2], Liang Ming[2], Ting Yan[2],
Richard Penty[1] & Qixiang Cheng [1,2] ✉

Miniaturized spectrometers hold great promise for in situ, in vitro, and even in vivo sensing applications. However, their size reduction imposes vital performance constraints in meeting the rigorous demands of spectroscopy, including fine resolution, high accuracy, and ultra-wide observation window. The prevailing view in the community holds that miniaturized spectrometers are most suitable for coarse identification of signature peaks. Here, we present an integrated reconstructive spectrometer that enables near-infrared (NIR) spectroscopic metrology, and demonstrate a fully packaged sensor with auxiliary electronics. Such a sensor operates over a 520 nm bandwidth together with a resolution below 8 pm, yielding a record-breaking bandwidth-to-resolution ratio of over 65,000. The classification of different types of solid substances and the concentration measurement of aqueous and organic solutions are performed, all achieving approximately 100% accuracy. Notably, the detection limit of our sensor matches that of commercial benchtop counterparts, which is as low as 0.1% (i.e. 100 mg/dL) for identifying the concentration of glucose solution.

Near-infrared (NIR; 780–2500 nm) spectrometry, one of the most essential vibrational spectroscopy techniques, is widely applied in numerous fields such as biomedicine, chemistry, and material science. Over the past decade, the surging demand for in situ, in vitro, and in vivo NIR spectroscopic analysis, including wearable devices for healthcare monitoring, portable tools for chemical detection, and compact optical systems for hyperspectral imaging, has driven the development of miniaturized spectrometers towards smaller sizes and higher performance. Leveraging the compatibility with mature CMOS technologies, silicon photonics offers a low-cost platform for developing chip-size spectrometers[1]. Additionally, the broad transparency windows of Si (from 1.1 to 8 μm) and SiN (from 370 to 5 μm) perfectly cover the spectral range for NIR detection[2]. Currently, most on-chip spectrometer designs adhere to the same principles as traditional dispersive or Fourier transform (FT) spectrometers but leverage photonic integration technologies to obtain a millimeter scale footprint[3]. The size reduction, however, in return sets limitations to the dispersive or interferometric elements, which fundamentally bound their bandwidth and resolution[4,5]. In recent years, the emergence of reconstructive spectrometers (RSs) with compressive sensing algorithms has paved a new way for miniaturized spectrometers[6]. In principle, RSs employ a global sampling strategy to resolve the entire incident spectrum, generally by forming an underdetermined matrix for extensive information acquisition. This feature makes them appealing to be implemented on chip, given their simplicity. However, the notable dispersion effects in the integration platform impose substantial constraints, especially on the operational bandwidth, either for passive devices[7–9], or active designs[10–13].

In contrast, NIR spectroscopy applications necessitate fine resolution, high accuracy, and wide working bandwidth for capturing informative spectral features[14]. Figure 1a summarizes the overtone absorption bands of typical functional groups in the NIR range[15]. The results clearly show that: (1) a functional group typically exhibits multiple overtone bands that span over hundreds of nanometers, and

---

[1]Electrical Engineering Division, Department of Engineering, University of Cambridge, Cambridge, UK. [2]GlitterinTech Limited, Xuzhou, China. [3]These authors contributed equally: Chunhui Yao, Wanlu Zhang. ✉e-mail: qc223@cam.ac.uk

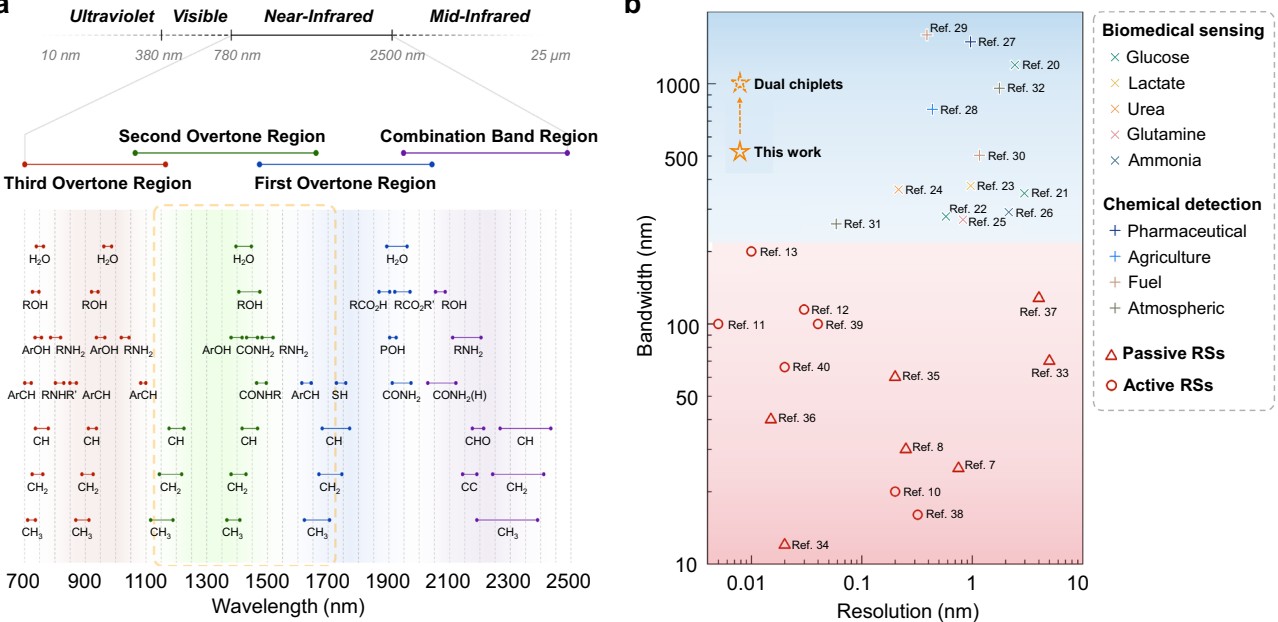

**Fig. 1 | Near-infrared spectroscopy needs vs. reconstructive spectrometer performance. a** Overtone absorption bands of typical functional groups in the near-infrared spectrum range. **b** Performance requirements of various near-infrared spectroscopy applications compared with the metrics of state-of-the-art waveguide-based reconstructive spectrometers. The colored dots represent specific application scenarios. The dashed arrow indicates that sensor performance can be further enhanced by co-packaging two or even more chiplets (see "Discussion" for details).

(2) the overtone bands of different functional groups overlap. The above attributes could largely complicate spectroscopic analysis, due to potential data underfitting[16]. For instance, the metrological measurement of typical biomarkers such as glucose, lactate, and urea, demands tracking various organic functional groups, including -OH, -CH, and -CH$_2$, thus requiring an ultra-wide observation window of hundreds of nanometers[17,18]. Likewise, the spectroscopic analysis of solid, liquid or gas substances for pharmaceutical, agriculture, fuel, and atmospheric monitoring may even need a functional bandwidth of over a thousand nanometers[19]. Figure 1b summarizes the bandwidth and resolution specifications adopted in various NIR applications for biomedical sensing[20–26] and industrial chemical detection[27–32], along with the performance metrics of state-of-the-art chip-scale RSs[7,8,10–13,33–40], where there remains a clear gap. It is thus commonly accepted that miniaturized spectrometers are more for the identification of signature spectral peaks rather than quantitative metrology[41].

In this work, we present an ultra-high-performance integrated RS that empowers NIR spectroscopic metrology. Our design simply consists of a cascade of tunable micro-ring resonators (MRRs) equipped with curved directional couplers (DCs). These MRRs are dispersion-engineered to operate under an over coupling state by balancing their wavelength-dependent round-trip loss and coupling efficiency, thereby creating efficient sampling responses across an ultra-wide wavelength range. Meanwhile, the sampling response is temporally decorrelated by manipulating the phase of each MRR. The RS chip is implemented on a SiN integration platform, and gets fully packaged into a chip-scale sensor with auxiliary electronics, demonstrating an over 520 nm operational bandwidth from 1180 to 1700 nm, as well as a superior resolution of below 8 pm. This corresponds to a bandwidth-to-resolution ratio exceeding 65,000, which, to the best of our knowledge, is significantly higher than any reported miniaturized spectrometer. A series of NIR spectroscopic applications is demonstrated, including the classification of plastic and coffee samples and the concentration measurement of aqueous and organic solutions, all achieving approximately 100% accuracy. Most importantly, the detection limit of our sensor is examined using glucose solution with concentrations as low as 0.1% (i.e. 100 mg/dL or 5.55 mmol/L) identified. Such level of detectability is already comparable to that of commercial benchtop spectrometers, establishing a new benchmark for NIR spectroscopy with miniaturized sensors.

## Results

### Principle and design

The working principle of RSs is elaborated in "Methods". As revealed by the compressive sensing theory, an RS necessitates a sufficient number of sampling channels with rapid and random spectral perturbations, also known as spectral speckles[42]. To achieve this, we propose an ultra-broadband single-bus RS based on a cascade of dispersion engineered MRRs, as illustrated by Fig. 2a. By opting for the SiN platform, our device benefits from a material dispersion that is over four times lower than that of the Si-on-insulator (SOI)[2]. Figure 2b presents the MRR's structural diagram in a vertical racetrack style. The transmission spectrum of such an MRR can be generally written as[43]:

$$T_i = \frac{\alpha^2 + r^2 - 2\alpha r \cos(\varphi_i + \delta_i)}{1 + \alpha^2 r^2 - 2\alpha r \cos(\varphi_i + \delta_i)} \quad (1)$$

where $r$ is the self-coupling coefficient, $\alpha$ denotes the loss coefficient, $\varphi_i$ is the single-pass phase shift of the ring, and $\delta_i$ represents the external phase tuning. Based on Eq. 1, the transmission intensity at resonance wavelength $T_{res}$ can be derived as:

$$T_{res} = \frac{(r - \alpha)^2}{(1 - r\alpha)^2} \quad (2)$$

Instead of operating under the stringent critical coupling condition as high-Q narrowband filters, the MRRs in our design are tailored to be over-coupled, targeting to feature resonance peaks with large full widths at half maximum (FWHMs) and moderate extinction ratios (i.e.

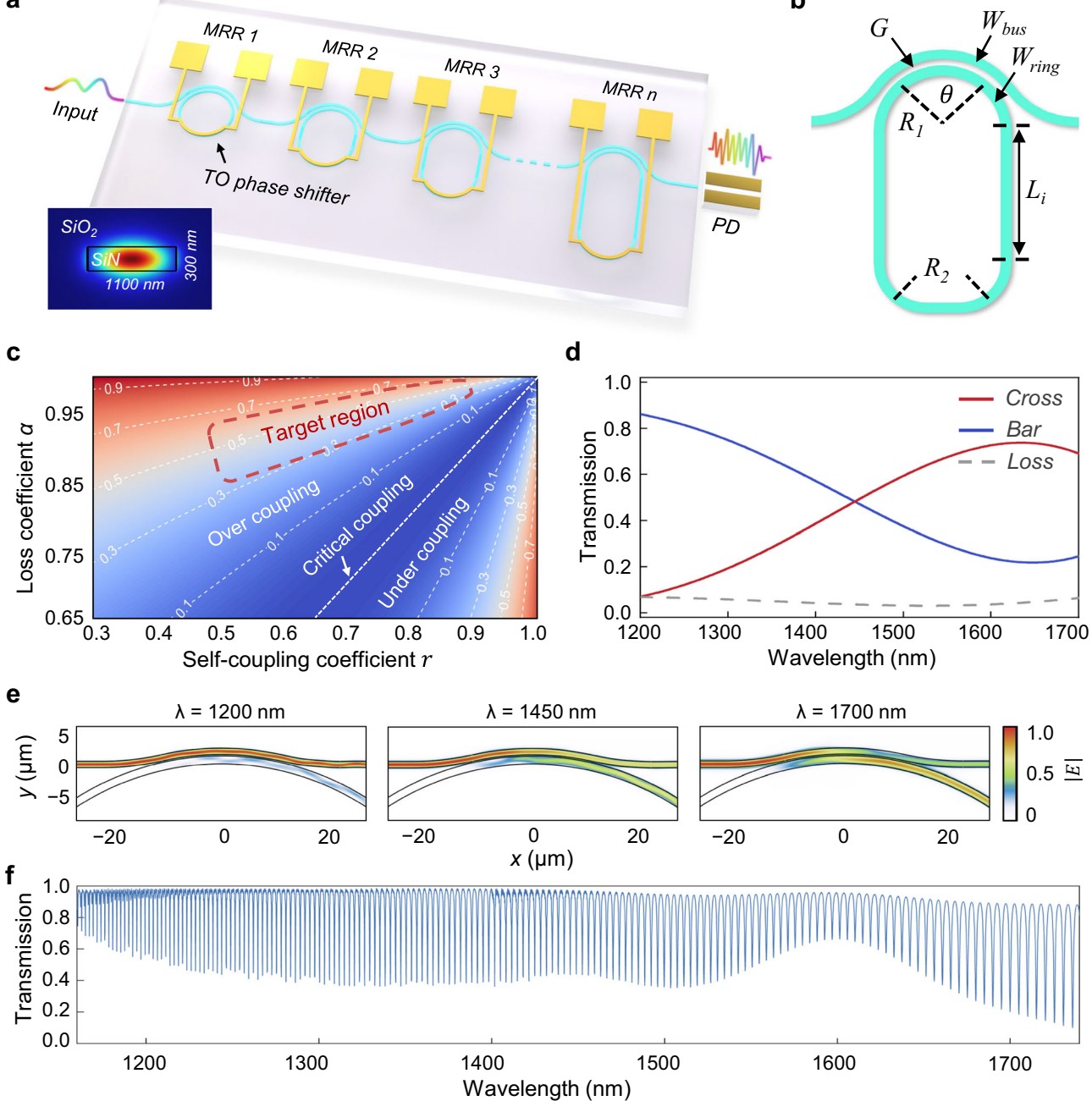

**Fig. 2 | Spectrometer design and simulations. a** Conceptual Schematic of the proposed ultra-broadband spectrometer featuring multiple stages of micro-ring resonators on a single bus. The inset shows the mode field distribution on the bus waveguide. **b** Structural diagram of a dispersion engineered ring resonators with a curved directional coupler. **c** Calculated values of $T_{res}$ under different combinations of $\alpha$ and $r$. The insets highlight the target region of $\alpha$ and $r$ for the micro-ring resonators. **d** Simulated coupling efficiency of the optimized curved directional coupler, showing an increasing cross-coupling efficiency over wavelength to match the rising round-trip loss on the ring. **e** Simulated light propagation profiles in the curved directional coupler at different wavelengths, shown as normalized electric field intensities. **f** Simulated spectral response for one of the cascaded ring resonators, showing the targeted over coupling state across an ultra-wide wavelength range.

moderate values of $T_{res}$). This yields substantial spectral perturbations in the overlaid response to ensure high sampling efficiency, while introducing minimal excess loss. Figure 2c shows the calculated $T_{res}$ under different combinations of $\alpha$ and $r$. The inset highlights the broad design space where an MRR can sustain an effective over-coupling state, while keeping the $T_{res}$ in a preferable range. Therefore, the key of our MRR design lies in maintaining a balance between the loss and coupling efficiency across a widest possible wavelength range.

For this purpose, we strategically deploy a pair of waveguide bends with a fixed but relatively small radius ($R_2$) to ensure the bending

loss dominate the losses on the ring, so that the round-trip loss naturally increases with longer wavelengths. Meanwhile, we adopt a curved DC structure for greater wavelength stability, thanks to its enhanced phase-matching capability[44]. Its structural parameters are globally optimized using the particle swarm optimization (PSO) algorithm to obtain a gradually increasing cross-coupling efficiency over wavelength, thereby matching the rising on-ring loss. Detailed design process is described in "Methods". Figure 2d, e presents the finite-difference time-domain (FDTD) simulated coupling efficiency and light propagation profiles of the tailored curved DC, respectively, showing a

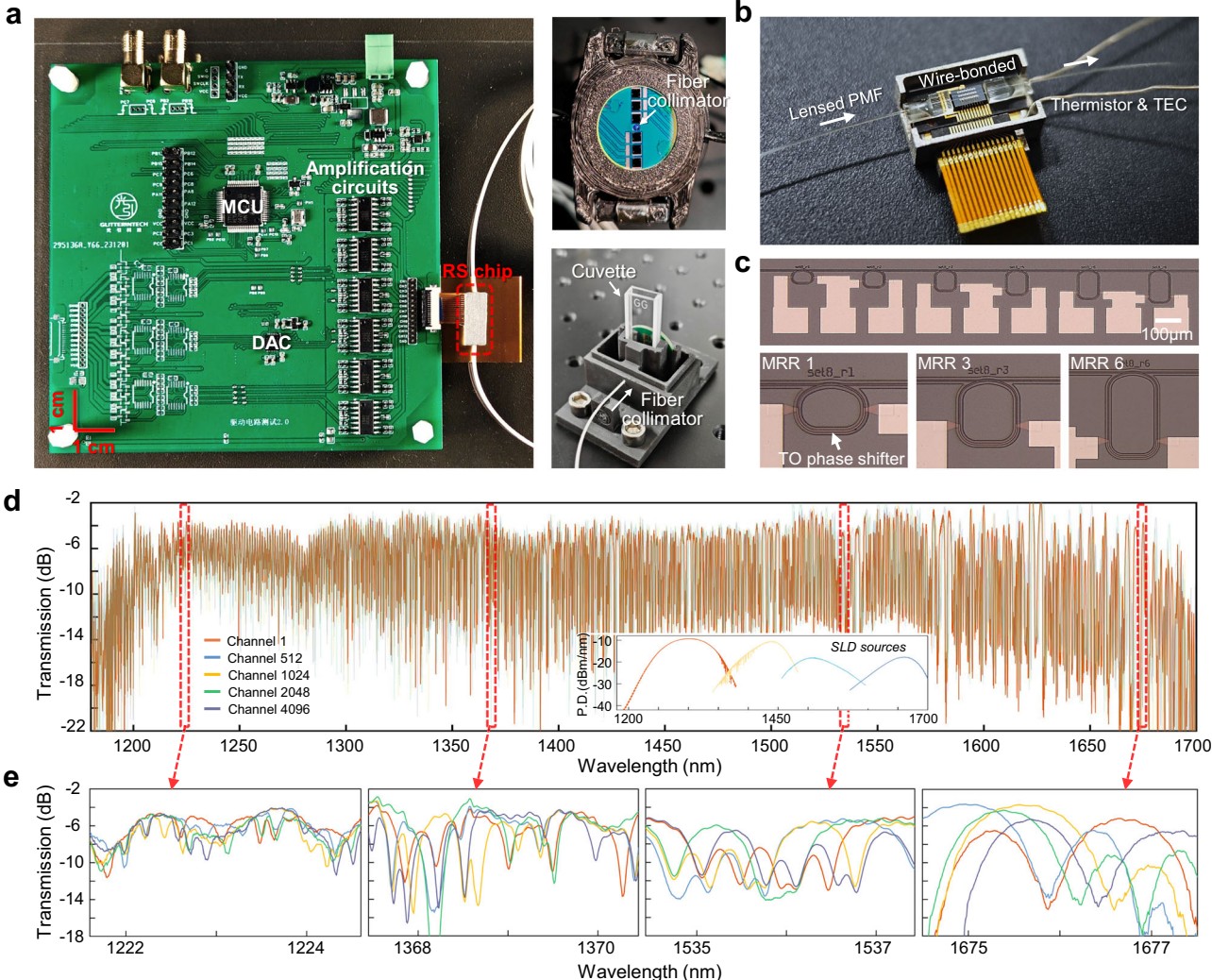

**Fig. 3 | Device images and measured channel spectral responses. a** Photograph of the miniaturized near-infrared spectrometric sensor. The insets show the optical sampling interfaces for measuring reflection and transmission spectra, respectively. MCU microcontroller unit. DAC digital-to-analog converter. **b** Enlarged image of the fully packaged spectrometer chip. PMF polarization-maintaining fiber. TEC thermoelectric cooler. **c** Microscope image of the fabricated spectrometer with six stages of tunable micro-ring resonators. The insets provide enlarged views of MRRs with different circumferences. TO: thermo-optic. **d** Representative examples of the measured channel spectral responses between 1180 nm and 1700 nm. The inset depicts the emission spectra of the four superluminescent diode (SLD) sources. P.D.: power density. **e** Channel responses at different observation windows.

rising cross-coupling ratio over wavelength. Besides, each MRR is equipped with a pair of straight waveguides with varying lengths ($L_i$) to not only achieve small FSRs for rapid spectral roll-offs, but also to break any periodicities in the overlaid response. For example, Fig. 2f plots the FDTD simulated spectral response for one of the cascaded MRRs, demonstrating that the desired over-coupling state is well maintained over an ultra-broad wavelength range from around 1160 to 1740 nm. Additionally, the MRRs incorporate thermo-optic (TO) phase shifters to temporally decorrelate the sampling responses. By tuning the MRR's phase (i.e. $\delta_i$) into multiple states between 0 and $2\pi$, the cascading system yields an exponentially scalable number of sampling channels, which equals to the cumulative product of phase states at each MRR. This design, therefore, effectively produces spectral sampling matrices with thousands of temporal channels.

### Experimental characterization
Figure 3a presents our fully packaged NIR spectroscopic sensor at centimeters scale, incorporating the SiN RS chip and a high-speed driving board with a microcontroller unit (MCU) integrated. The insets enlarge two optical sampling interfaces that are tailored for reflective and transmissive measurements, respectively. More details regarding the driving board and sampling interfaces are provided in "Methods". The RS chip is designed to incorporate six cascaded MRRs as a balanced choice between device performance, system footprint/complexity, and the consumption of sampling channels. Figure 3b details the device packaging, where the chip is wire-bonded for electrical fanout and optically accessed via lensed polarization-maintaining fibers (PMFs). A microscope image of the RS chip is shown in Fig. 3c, with the insets magnifying three tunable MRRs with curved DC. The MRRs each occupies a footprint of less than $80 \times 150\,\mu m^2$ and are laterally spaced by 200 μm to minimize thermal crosstalk. We set four phase states per MRR, creating a total of 4096 temporal sampling channels. The power consumption of each MRR experimentally measures around 34 mW/rad, resulting in an average system power consumption of 240 mW. This can be further enhanced by incorporating deep trenches or undercuts in the waveguide to improve thermal efficiency. For the calibration of RS chip, four superluminescent diodes (SLDs) centered at different wavelengths are

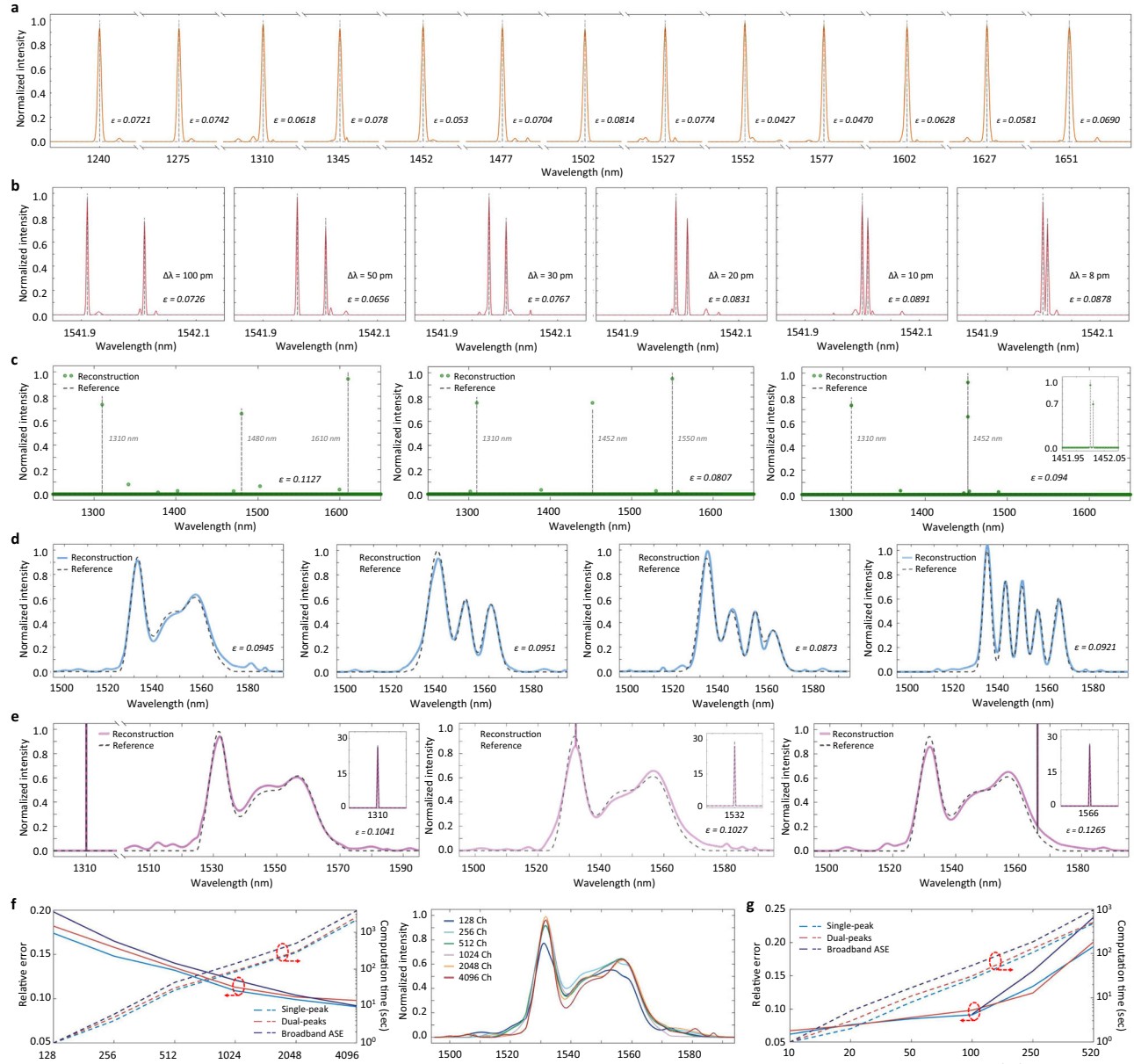

**Fig. 4 | Performance characterization of the RS. a** Reconstructed spectra for a series of single-peak laser signals. The black dashed lines indicating their center wavelengths. **b** Reconstructed spectra for dual-peak laser signals with decreasing spectral spacing from 100 pm down to 8 pm. **c** Reconstructed spectra for triple-peak laser signals at different wavelength positions, with the spectral spacings between peaks ranging from as close as 10 pm to over 200 nm. **d** Reconstructed broadband signals generated by an Erbium-doped fiber amplifier, filtered by a commercial waveshaper with various spectral patterns. **e** Reconstructed spectra of hybrid signals containing both broadband and narrowband spectral components.

The inset shows the reconstruction of the narrowband laser peaks at different wavelengths. **f** Left: investigation of reconstruction accuracy and computing time versus sampling channel number. The solid lines represent the reconstruction accuracies, while the dashed lines denote the computational costs. Right: the reconstructed broadband spectra using different numbers of sampling channels. **g** Relationship between the reconstruction accuracy and computing time versus the width of reconstruction window for various types of input signals (with channel number fixed at 512). Note that the minimal window width for the broadband signal is restricted to 100 nm to prevent any information loss.

introduced as light sources (see more discussions in "Methods"). Figure 3d plots representative measured sampling channels over a 520 nm bandwidth between 1180 and 1700 nm (Supplementary Fig. S1 shows the whole sampling matrix). The inset shows the emission spectra of the four SLDs. Figure 3e further displays the channel responses at four different observation windows, highlighting the random spectral fluctuations.

To characterize the RS performance, we first test a variety of discrete incident spectra, including single-, dual-, and triple-peak laser signals at different wavelengths. The reconstruction algorithm and

processes are detailed in "Methods". The reconstruction accuracies are quantified by the *L2-norm* relative errors $\varepsilon$, defined as $\varepsilon = ||\Phi_0 - \Phi||_2 / ||\Phi_0||_2$ where $\Phi_0$ and $\Phi$ denote the reference and reconstructed spectra, respectively. As shown in Fig. 4a–c, our RS precisely resolves the intensities and locations for all peaks, exhibiting low relative errors ranging from 0.04 to 0.11. The minor non-zero peaks apart from the laser signals can be attributed to the reconstruction errors caused by inevitable measurement inaccuracies, which can be further suppressed by enhancing the system's signal-to-noise ratio and applying more advanced reconstruction algorithms.

Notably, the spectral spacing of dual-peak signals is gradually reduced down to 8 pm, marking its resolution according to the Rayleigh criterion. Rigorous simulations further show that such a sensing resolution can be well achieved across the entire operational bandwidth (see Supplementary Fig. S2). Subsequently, various continuous, broadband spectra that are generated by a benchtop waveshaper are tested. Figure 4d depicts that all complex waveform features are well reconstructed. Hybrid incident spectra have also been measured, with different locations of a laser peak combining the amplified spontaneous emission (ASE) spectra of an Erbium-doped fiber amplifier (EDFA). Figure 4e shows that both the broadband and narrowband spectral components are well distinguished for all cases.

The global sampling feature of our RS offers the flexibility to customize the number of sampling channels in any individual measurement, allowing a trade-off between the reconstruction accuracy, measurement time, and computational complexity to enable user-definable performance metrics[13]. Accordingly, we investigate the impact of the channel number on reconstruction performance, covering all cases of single-peak, dual-peak, and broadband inputs. Here, the spectral resolution is scaled down in accordance with the decrease of sampling channels to maintain a consistent compression ratio. Figure 4f plots the reconstruction error and computing time as a function of the channel number (using a Xeon 10980 CPU with 64 GB of memory), and highlights the reconstructed broadband spectra under different channel numbers. As can be seen, while reducing the channel number gradually induces a higher relative error, it sharply lowers the computational cost. On the other hand, the impact of spectral window width is explored, as shown by Fig. 4g, revealing that reducing the redundant bandwidth in reconstruction leads to clear accuracy enhancement. This is attributed to the noise accumulated over the signal bandwidth that does not carry inputs (see Methods for

detailed strategies on adjusting the reconstruction window). Furthermore, we examine the temperature stability of our RS. Measurements reveal that the sampling response redshifts by only 0.32 nm when the ambient temperature increases from 10 to 30 °C, which can be easily handled by tweaking the reconstruction matrix and potentially enables a cooling-free operation. This is also translated into a temperature tolerance of around ± 0.75 °C (see Supplementary Fig. S3).

## Applications for NIR spectrometric sensing

With the fully-packaged NIR sensor, we showcase a series of bandwidth-demanding spectroscopic applications for material classification and solution concentration monitoring, addressing typical industrial requirements[1]. We examine ten different types of plastics commonly used in chemical manufacturing and ten varieties of coffee powers originated from diverse regions. These samples are visually indistinguishable, but exhibit subtle spectral differences over the NIR range. We repeatedly measure each sample for 60 times over the entire 520 nm bandwidth, and randomly split the data into training and testing sets in a 7:3 ratio. Note that for all these measurements, we downsize the sampling channel number to 312 as a balanced choice between computational cost and reconstruction performance (please see Fig. 4f, g), given that the spectra under test exhibit modest roll-off ratios. Figure 5a, b shows the measured reflectance of two representative samples, respectively, for the plastic and coffee. The reflectance spectra for all remaining samples are provided in Supplementary Fig. S4. The average spectral relative errors for these plastic and coffee samples range between 0.026 and 0.056, as shown by Supplementary Table S1. The minor discontinuities in the measured spectra are due to the low SLD power density at those wavelength regions (see Fig. 3d). Corresponding classification models are trained on the basis of support vector machine (SVM)[45] to identify plastic or coffee samples.

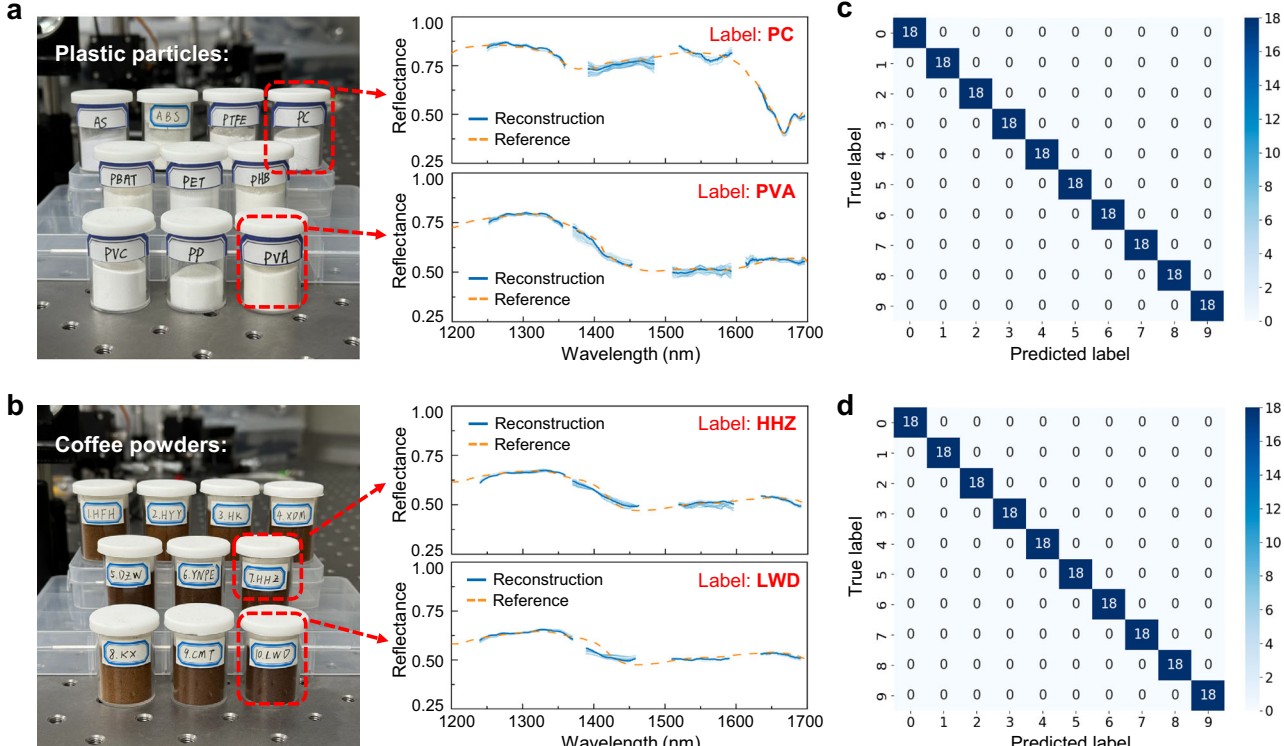

**Fig. 5 | Classification of various solid substances. a, b** Reflectance measurement of ten different types of plastic and coffee samples (labeled by various names), respectively. The insets show the measured reflectance spectra of two representative plastic or coffee samples (each repeated for 60 times), respectively, revealing superior measurement repeatability of our sensor. The orange dashed lines represent the references measured using a commercial benchtop spectrometer. One of the repetitions is highlighted to indicate the high reconstruction accuracy. **c, d** Confusion matrix of the classification results for various plastic or coffee samples, respectively.

Figure 5c, d presents the classification results for the plastic and coffee samples, respectively, all demonstrating 100% accuracy. A detailed investigation into the relationship between model prediction accuracy and reconstruction errors is also conducted via simulations, as shown by Supplementary Fig. S5. The analysis reveals that the 100% accuracy begins to decline when relative errors exceed certain thresholds, with turning points for plastic and coffee samples occurring at around 0.08 and 0.06, respectively.

The concentration monitoring for various aqueous and organic solutions is further performed. We test the ethanol aqueous solutions, solutions of ethylene glycol (EG) in isopropanol (ISO), and glucose aqueous solutions. Figure 6a shows the measured absorptance spectra of ethanol solutions, which, despite the strong water absorption bands, display the unique NIR features of ethanol. For example, from 1640 to 1700 nm, the measured absorptance first decreases and then increases as the ethanol concentration rises, with a turning point at around 1675 nm. This distinctive signature matches the results from reported studies[46], and can be explained by the absorption of the C-H bonds in ethanol (see Fig. 1a). Accordingly, we train a prediction model using the random forest (RF) algorithm (see "Methods" for modeling details)[47]. Here, the coefficient of determination ($R^2$) is used to assess model's fitting performance, defined as $R^2 = 1 - \frac{\sum_i (y_i - \hat{y}_i)^2}{\sum_i (y_i - \bar{y})^2}$, where $y_i$ and

$\hat{y}_i$ denote the true and predicted concentration value of sample $i$ in the testing set, respectively, and $\bar{y}$ represents the mean of the true value. As a result, our RF model achieves a coefficient of determination ($R^2$) of 1.000 on the testing sets, indicating an excellent fit. Figure 6b presents a scatter plot of the measured and predicted ethanol concentrations. The calculated mean absolute error (MAE) and root mean squared error (RMSE) for the testing sets are only 0.057% and 0.074%, respectively. Figure 6c presents the measured absorptance spectra for EG solutions in ISO, showing clear concentration-induced variations in absorption. It is worth noting that distinguishing EG from ISO is crucial, since EG can cause severe renal failure and is thus more hazardous than ISO[48,49]. For instance, across the broad wavelength range between around 1400 nm to 1680 nm, a significant rise in absorptance can be observed as the concentration of EG increases, which is attributed to that EG has one more -OH group than ISO. Figure 6d shows the prediction results using our RF model, with the MAE and RMSE being only 0.029% and 0.058%, respectively. The measured absorptance spectra of various glucose solutions are shown in Fig. 6e. As can be seen, the spectral variations are more pronounced in the wavelength ranges of below 1400 nm and above 1600 nm, where the water absorption is minor and the overtone vibrations of glucose molecule's C-H bonds dominate[50]. Figure 6f compares the actual glucose concentrations against the predictions from our model. The calculated MAE and RMSE

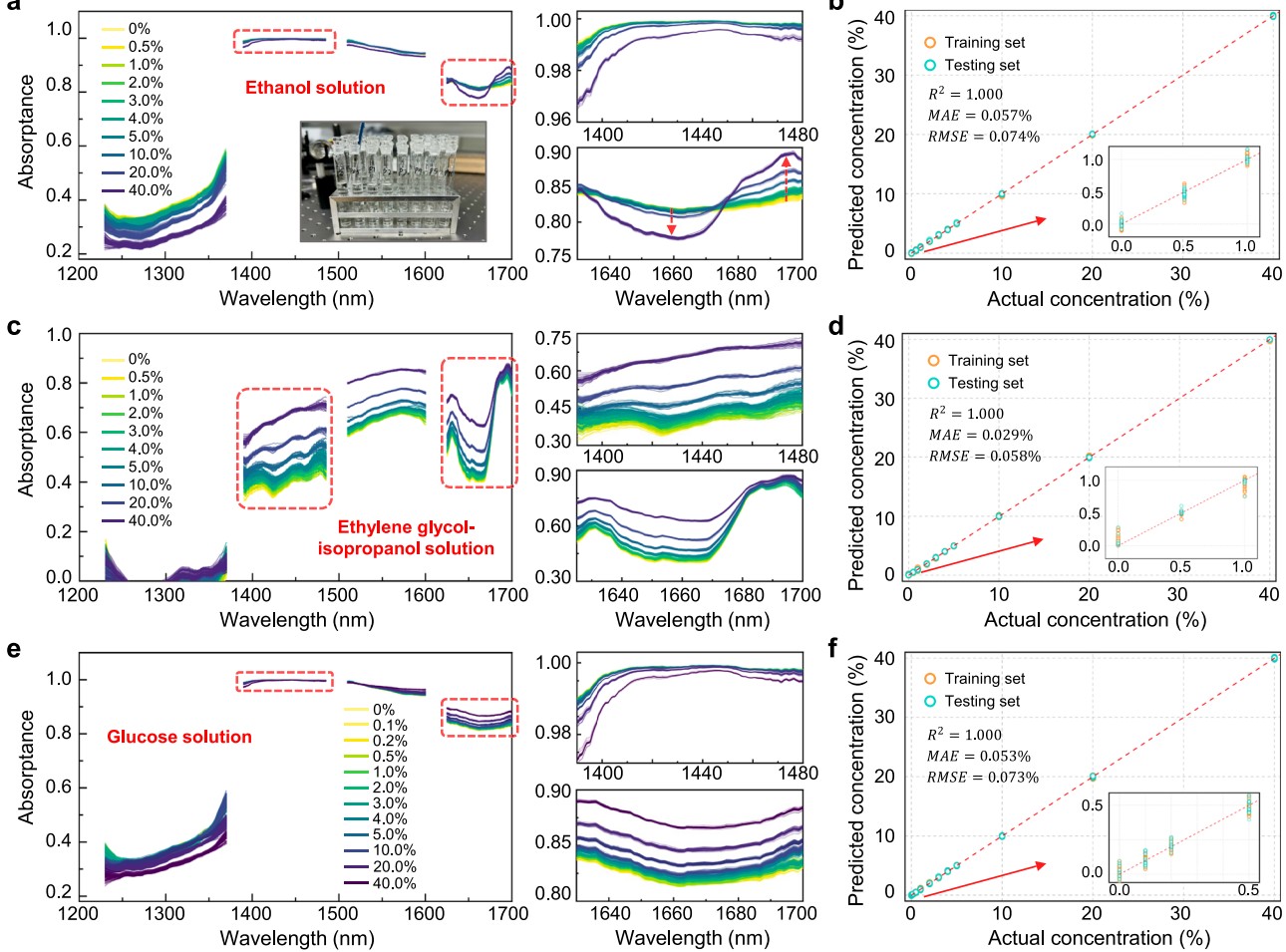

**Fig. 6 | Concentration testing of various solutions. a** Measured absorptance spectra for ethanol aqueous solutions with concentrations decreasing from 40 to 0.5% (each repeated for 60 times). The inset shows the test tubes containing the respective solutions. The red arrows denote the trend of absorptance variation with increasing solution concentration. **b** Predicted ethanol concentrations using our random forest model. The inset provides an enlarged view of the prediction results at low concentrations. **c** Measured absorptance spectra for solutions of ethylene glycol in isopropanol with concentrations decreasing from 40 to 0.5%. **d** Predicted concentrations using our random forest model. **e** Measured absorptance spectra for glucose aqueous solutions with concentrations decreasing from 40% down to 0.1%. **f** Predicted glucose concentrations using our random forest model.

**Table 1 | Performance comparison based on the concentration test of glucose solutions**

| Principle | Spectral window | Detection limit | MAE | RMSE |
|---|---|---|---|---|
| Dispersion (IdeaOptics NIR17+Px) | 900–1700 nm | 0.1% | 0.022% | 0.029% |
| Fourier transform (Bruker MPA II) | 866–2500 nm | 0.05% | 0.002% | 0.011% |
| Dispersion (IdeaOptics NIR17+Px) | 1180–1700 nm | 0.1% | 0.026% | 0.035% |
| Fourier transform (Bruker MPA II) | 1180–1700 nm | 0.1% | 0.013% | 0.024% |
| Reconstruction (this work) | 1180–1700 nm | 0.1% | 0.053% | 0.073% |

are 0.053% and 0.073%, respectively, illustrating the system detection limit of at-least 0.1% (i.e. 100 mg/dL or 5.55 mmol/L).

## Discussion

We replicate the glucose concentration tests using benchtop dispersive and FT spectrometers (IdeaOptics NIR17+Px and Bruker MPA II), in tandem with commercial broadband NIR sources based on tungsten-halogen lamps. Note that here we construct different prediction models using their full or partial operational bandwidth for fair comparisons (see Supplementary Fig. S6 for more details). Table 1 details that our NIR sensor reaches the detection limit and accuracy comparable to those of commercial counterparts. The relatively higher MAE and RMSE can be attributed to the difference in measurement stability, which can be improved by optimizing the chip control system and employing a more stable broadband light source.

Since only MRRs are used to shape the channel sampling response, our design allows the flexible migration of operational wavelength band. For instance, MRRs with dispersion-engineered DC targeting the waveguide range of between 700 and 1200 nm are readily optimized, as shown by Supplementary Fig. S7. Hence, simply by co-packaging two or more RS chiplets, a miniaturized NIR sensor with over one thousand nanometer bandwidth can be easily achieved.

In summary, we present an RS-based NIR spectroscopic sensor, achieving an >520 nm operational bandwidth with a <8 pm resolution. Various spectroscopic applications are examined, achieving accuracies of approximately 100%. We also demonstrate the detection limit of our sensor to be at-least 0.1% (i.e. 100 mg/dL), which is comparable to the results obtained from commercial benchtop spectrometers, representing a big leap towards spectroscopic metrology with miniaturized spectrometers.

## Methods

### Parameter optimization of curved DC

We utilize ANSYS Lumerical FDTD to perform the physical simulations of the curved DC and employ a custom PSO algorithm script to globally optimize the DC geometry[51]. In specific, we first conduct a coarse sweep of all structural parameters to identify a relatively optimal combination, which serves as the initial parameter set. The power intensities at both output ports of the DC are used to calculate the objective function, aiming to achieve a gradually increasing cross-coupling efficiency over the widest possible wavelength range while maintaining a minimal insertion loss. Based on this, the PSO algorithm continuously iterates to obtain the optimal parameter set. The detailed structural parameters of our curved DC are listed in Supplementary Table S2.

### Chip fabrication and packaging

The spectrometer chip is fabricated via a CORNERSTONE SiN multi-project wafer run, employing standard deep ultraviolet (DUV) lithography with a feature size of 250 nm. It consists of a 300 nm thick SiN layer sandwiched between a 3 μm buried oxide layer and a 2 μm Silicon dioxide top cladding layer. The chip is diced, polished, and wire-bonded to customized PCB broad for electrical fan-out (see Fig. 3b). Lensed PMFs are used to match the waveguide edge couplers on both edges for efficient optical assessment. UV-curable adhesive is applied to mechanically secure the PMFs for stable optical alignment, measuring a coupling loss of around 2.5 dB per facet.

### Electrical control and thermal stabilization

An automatic electrical driving board is developed to enable the swift modulation of channel sampling responses, as shown by Fig. 3a. The MCU is programmed to produce the initial control signals and send them to a high-resolution multi-channel digital-to-analog converter (DAC). The output analog signals from the DAC are then amplified by amplification circuits and injected into the spectrometer chip for temporal channel sweeping. This system allows for a sampling speed of over 500 Hz, such that the measurement for each spectrum can be completed within a few seconds.

A thermistor is attached to the top surface of the chip, while a thermoelectric cooler (TEC) is placed underneath it (see Fig. 3b), jointly forming a negative feedback mechanism to facilitate the temperature stabilization during experimental testing.

### Optical testbed and sampling interfaces

To calibrate our ultra-broadband RS chip, we sequentially introduce four SLDs with different center wavelengths as inputs and measure the corresponding output spectra using a commercial benchtop spectrum analyzer. The results are then compiled to form the full-band spectral response. Likewise, during the NIR spectrometric sensing for various substances, these SLDs are also sequentially turned on to illuminate the samples. The reflected or transmitted power is then collected by photodiodes (PDs) and used for spectral reconstruction. After normalization with standard references (e.g., a fully reflective whiteboard or an unloaded cuvette), the reflectance or absorptance of the samples across the full bandwidth can be obtained. Supplementary Fig. S8 details the schematic of the testbed workflow.

To efficiently collect the reflected or transmitted light from samples, we develop two free-space optical sampling interfaces, as shown in the insets in Fig. 3a. Supplementary Fig. S9 details their structure and working principles, respectively. The first interface measures the reflectance of solid substances using a single-mode fiber collimator with multiple 2 mm photodiodes (PDs). The sample is positioned above the collimator, ensuring the emitted light reaches and reflects off the sample before being collected by the PDs This process allows a coupling loss of less than 8 dB. The second interface is designed for measuring the absorptance of liquid samples, where the single-mode fiber collimator is aligned with a cuvette that contains the sample. The transmitted light through the solution is captured using surface PDs. When the cuvette is empty, this interface exhibits a coupling loss of less than 1 dB.

### Reconstruction algorithm and process

Mathematically, the underlying principle of RSs can be described using an undetermined equation, written as[1,41]:

$$I_{M \times 1} = S_{M \times N} \Phi_{N \times 1} \tag{3}$$

where $\Phi_{N \times 1}$ denotes the discretized vector of an unknown incident spectrum with $N$ spectral pixels, $S_{M \times N}$ represents the sampling matrix with a channel number of $M$, and $I_{M \times 1}$ is the corresponding output power intensities for each sampling channel. Using regression algorithms, the incident spectrum $\Phi$ can be accurately retrieved even when $M$ is considerably smaller than $N$, following:

$$\text{Minimize } ||I - S\Phi||_2 \text{ subject to } 0 \leq \Phi \leq 1 \tag{4}$$

Equation (4) is suitable for solving discrete signals (e.g., laser signals). For continuous signals, additional regularization terms can be added to mitigate the ill-conditioning of the undetermined problem, such as the modified Tikhonov regularization[52], modifying Eq. (4) to:

$$\text{Minimize } ||I - S\Phi||_2 + \alpha||\Gamma\Phi||_2 \text{ subject to } 0 \leq \Phi \leq 1 \qquad (5)$$

where $\Gamma$ is a difference-operator that calculates the derivative of $\Phi$, and $\alpha$ is the regularization weight that helps suppress noise-induced reconstruction errors. For more complex hybrid incidence (i.e. with both discrete and continuous signals), the Eq. (5) should be further modified by introducing segmented regularization terms, as:

$$\text{Minimize } ||I - T(\Phi_1 + \Phi_2)||_2 + \alpha||\Phi_1||_1 + \beta||\Gamma_2\Phi_2||_2 \text{ subject to } 0 \leq \Phi_1, \Phi_2 \leq 1 \qquad (6)$$

where $\Phi_1$ and $\Phi_2$ denote the narrowband and broadband spectral components, respectively; $\alpha$ and $\beta$ are the corresponding weight coefficients. In our study, all spectra reconstruction are performed based on the above equations using the CVX regression algorithm[53].

While our RS demonstrates ultra-high resolution and bandwidth, it is not always suitable to perform every reconstruction using its full capabilities considering not only the computational burden (see Fig. 4f, g), but also the fact that practical incident signals usually only occupy a certain spectral range. Therefore, we adopt a progressive strategy for spectral reconstruction, starting with a coarse reconstruction over the entire bandwidth and moving to a fine reconstruction within a selected spectral window. Specifically, an initial low-resolution, full-band reconstruction is conducted to locate the main energy distribution of the unknown incident spectrum. After identifying its general spectral range, we then narrow the reconstruction window and increase the resolution until the optimal result is achieved. On the other hand, prior knowledge about the incident spectra can be also used to enhance the reconstruction efficiency. For instance, in our NIR sensing experiments, all samples are sequentially illuminated by a group of SLDs, which allows us to adjust the reconstruction window according to the coverage of the corresponding SLD.

## Sample preparation
The solid samples used in the experiments include ten different types of plastic and coffee. The ten plastic samples cover a range of commonly-used polymer and copolymer materials with different chemical compositions and physical properties, such as polycarbonate (PC), polyvinyl alcohol (PVA), acrylonitrile styrene (AS), and others. The ten coffee samples are produced from various regions, such as East Java, Rwanda, and Yunnan, and also vary in tree species and roasting methods. During our testing, all samples are ground and sieved to ensure uniformity in particle size. The processed particles are then placed into glass containers for reflectance testing.

The liquid samples include the aqueous solutions of glucose and ethanol, and the organic solutions of EG in ISO, all carefully prepared at varying concentrations. The glucose solutions are made by mass percentage, while the ethanol solutions and EG solutions are prepared by volume percentage. The solutions are then transferred into cuvettes, sealed, and tested for absorptance.

## Modeling of NIR spectrometric sensing
For the reflectance spectra of solid samples, we employ the SVM algorithm to develop classification models. SVM is a supervised learning technique that finds the optimal hyperplane in the feature space to separate data points of different classes. The model performance is evaluated based on classification accuracy, with the model's hyperparameters tuned to achieve optimal average results in cross-validation on the training set, thereby minimizing the risk of overfitting. The trained model is then used to classify the plastic samples in the testing set.

Regarding the absorptance spectra of solution samples, RF algorithm is adopted to train prediction models. RF is an ensemble learning method that constructs multiple decision trees to make predictions, where each tree is trained on random subsets of the data to enhance overall predictive performance. Here, we employ the random forest regressor from the Scikit-Learn library in Python for model construction. RMSE is used as the criterion to optimize model's key hyperparameters, such as the number of decision trees and the maximum depth of each tree. By fine-tuning these hyperparameters, we ensure a balance between model complexity and generalization. The optimized model is then applied to predict the concentrations of test samples.

## Reporting summary
Further information on research design is available in the Nature Portfolio Reporting Summary linked to this article.

## Data availability
All data needed to evaluate the conclusions in the paper are present in the paper and/or the Supplementary Materials. A collection of the spectra data in Fig. 6 is available in the University of Cambridge Repository at https://doi.org/10.17863/CAM.110905. Additional inquiries regarding the data can be directed to the corresponding author.

## Code availability
The spectra reconstruction algorithm, i.e. the CVX convex optimization tool, is available at: https://cvxr.com/cvx/. The code for the SVM and RF models is available from the corresponding author upon request.

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

## Acknowledgements

This research was supported by the UK EPSRC through project QUDOS (EP/T028475/1) and the European Union's Horizon 2020 Research and Innovation program through project INSPIRE (101017088), and also received support from GlitterinTech Limited. The authors thank Mr. Tao Zhang, Mr. Bobo Liu and Ms. Mengting Wu, Mr. Yanlong Liang, and Mr. Yahang Chen for the help in experiments.

## Author contributions

C.Y. conceived the spectrometer design, performed the optical simulations (with P.B.'s contribution), and drawn the chip layout. W.Zhang performed the characterization of spectrometer performance and analyzed the data with C.Y.'s help. C.Y. designed and conducted the NIR spectroscopic sensing experiments with J.M., W.Zhuo, J.Z., L.M. and T.Y.'s assistance. J.M., M.C., and T.Y. developed the electrical driving broads. C.Y. drafted the manuscript, with W.Zhang, Z.S., Y.Y. and Q.C.'s input. R.P. and Q.C. supervised the project.

## Competing interests

GlitterinTech Limited declares a pending patent application filed with the China National Intellectual Property Administration (inventors: C.Y. and Q.C., application number: CN2023107701923), pertaining to the spectrometer design presented in this manuscript. The authors declare no other competing interests.
