## [Transparent Peer Review file · Nature Communications]

Chip-scale sensor for spectroscopic metrology

Corresponding Author: Dr Qixiang Cheng

Version 0:

Reviewer comments:

Reviewer #1

(Remarks to the Author)

This manuscript presents a fully packaged sensing chip for spectroscopic metrology. The authors point out three requirements for current spectral analysis applications: high resolution, high accuracy and ultra-wide bandwidth. The spectrometer proposed in the manuscript achieves high performance with a cascaded custom MRR, and the results of two applications tested by the authors are also convincing. But I think there are still some issues that need to be addressed before considering publishing this manuscript:

1. The idea of a cascade MRR presented in this manuscript seems to be similar to that of the authors' previous paper [Ref.1], can the authors explain what innovative breakthroughs are made in the design of this manuscript compared to the previous ones.
2. It is mentioned that the MRR is designed to be over-coupled, but the reason is not given, and it is suggested to elaborate in more detail.
3. I don't quite understand the reason why line 103 mentioned that the round-trip loss increases with wavelength
4. In Figure 2(e), most of the short-wavelength light remains in the bus, and only a small amount is coupled to the micro-ring, so how to effectively decouple the light?
5. In Figure 4(f), the calculation time of the vertical axis is the result after normalization. Could the authors provide the order of magnitude for the computation time?
6. Image clarity is relatively low. The color distinctions between different curves in the right-side image of Figure 4(f) are not apparent.
7. What is the power consumption of this structure, given that the thermo-optic coefficient of the SiN material is relatively low?
8. In line 174, it is mentioned that the plastic particles' spectra have a moderate roll-off ratios, so only 312 sampling channels were used. Does this mean that the higher the roll-off ratio of an unknown spectrum, the faster the corresponding spectral fluctuations, and thus more sampling channels are required?
9. The determination coefficient of random forest algorithm is suggested to be explained.
10. In the method, the authors mention that when designing the DC, the coupling efficiency should gradually increase with wavelength. What is the purpose of this approach?

Reference

1. Yao, C. et al. Integrated reconstructive spectrometer with programmable photonic circuits. Nat. Commun. 14, 6376 (2023).

Reviewer #2

(Remarks to the Author)

In the paper, the authors develop a miniature NIR reconstructive spectrometer with 520nm bandwidth and resolution of 8pm. They use this device to demonstrate the identification of several chemicals. Some key innovations are, the use of SiN instead of SIO to reduce material dispersion, moderate extinction ratios for resonators to reduce loss, and using random variation in the free spectral ranges of the resonators to increase randomness to allow decorrelation of different channels. Overall, the resolution, bandwidth, and size of this device make this an important step forward for the community.

Scientific Inquiries

- 1) Are the authors confident in their claim of 520 nm bandwidth for their design? The data presented in Figure 3 d shows severe noise at 1670 nm and 1200 nm, this would reduce the operating bandwidth to 470 nm, a roughly 10% decrease from the original claim. Can the authors provide some evidence for the full spectral range of their device and explain how they are able to overcome the high noise at these wavelengths to discern the 8pm resolution?
- 2) The authors have chosen to use 6 MRRs in their design, can the authors discuss the reason for this design choice in more detail?
- 3) There is some evidence of spectral reconstruction provided in Figure 4a and b using lasers as inputs. All the peaks presented here have side lobes or peaks of notable intensity at adjacent wavelengths to the main peaks. Can the authors specify whether these peaks are an artifact of the input light, so an artifact of the device or reconstruction algorithm.
- 4) The authors should provide some more details on how the computation cost in their Figures 4f and g are calculated. Is this a computational time metric or number of units required etc.?
- 5) While the spectral reconstruction is quite accurate, there is a consistent underestimation of the signal at ~1530-1535 nm for Figures 4d and e. The consistency of this error is puzzling. Can the authors comment on whether this is a system artifact?
- 6) In Figure 5, the reconstructed spectra are provided along with the reference spectra of various samples. While the spectra match reasonably well, it would be useful to have a quantitative error measurement with each of the presented samples. Also, are the authors able to provide the maximum allowable error for their algorithm to predict the sample correctly? These two metrics will help distinguish the fidelity of the device from the predicting power of the algorithm.
- 7) While the detection limit and bandwidth of the presented device are comparable to commercial spectrometers according to Table 1, the mean absolute error and root mean squared error are substantially higher. Could the authors comment on the reason for this and the following? How is the spectrometer able to achieve similar detection limits while having higher errors? What is the limitation caused by this error?

Presentation Notes

- 8) In Figure 2e, the authors illustrate the simulated light paths through their device. Can the authors please add an annotation of the color bar to indicate the units depicted and whether this is simply a normalized scale with arbitrary units?
- 9) In Figure 5, it would be useful to provide the spectrum of the source in an inset to prevent confusion about the missing spectral bands Figure 5b. If possible, the authors are requested to provide some measurements with a broadband source to avoid this issue.

Version 1:

Reviewer comments:

Reviewer #1

(Remarks to the Author)

The authors addressed all the raised concerns and considerably improved quality of the manuscript. The submission can be accepted for publishing.

Reviewer #2

(Remarks to the Author)

In this paper, the authors focused on a computational on-chip approach to the traditional free-space optics spectrometer. Upon reviewing the author's responses, I have identified the following key changes in the manuscript in response to my prior critiques and comments.

1. The authors' have investigated the bandwidth issue in more detail and presented a more convincing demonstrating of their claimed >520 nm spectral bandwidth. They have provided FDTD simulations to confirm their observations and provided new data at the highest sensitivity of their equipment illustrate the bandwidth improvement. I believe this improvement provides sufficient scientific backing to the authors' bandwidth claims.
2. The revised manuscript also elaborates on the engineering decision to use 6 MRRs as a compromise between device footprint and resolution. The added analysis sufficiently clarifies this choice for scientists looking to replicate this group's results.
3. The authors are not able to provide a solution to the measurement inaccuracies I noted in their Figures showing some reconstructed spectra. They attribute these inaccuracies to reconstruction error from their algorithm. My opinion is that, by highlighting this shortcoming in their manuscript, the limitations of this technology are adequately described for the benefit of the community.
4. In the original manuscript, the authors used a nebulous metric defined simply as "computation" to represent the computational effort required for each configuration. However, I suggested using a better defined, more quantitative measure to allow comparison with other studies. A such, the authors have now adopted computation time as the metric for measuring computation effort. The specifics of their computational environment are also provided to allow researchers to compare and understand this metric.
5. The authors provide some more clarification of the inaccuracies and limitations of their system.

6. In the original report, the authors demonstrated their device by simply showing the predicted and measured spectral curves of their test samples. I recommended that the authors quantify the error between the spectrum as measured by their system and by a calibrated benchmark device. Taking this recommendation is accepted by the authors as they have provided new data with a quantification of the error in their spectra. I also suggested that the authors measure the maximum allowable error at which their prediction algorithm is able to recognize the material they are spectrally analyzing. The authors conducted new experiments to provide the readers with these numbers to establish the fidelity of their device.

7. I noted some key shortcomings of this system compared to conventional spectrometers, particularly the high mean absolute error and root mean square error in the measurements of the authors' device compared to commercial spectrometers. The authors have noted in the manuscript that this is a shortcoming of their approach as the on-chip approach is prone to repeatability error from measurement instability. While this reduces the attraction of the system, I still believe it is adequately novel and the authors' approach is satisfactory.

8. Some of the figures in the original manuscript were not annotated with a quantitative color bar, upon suggestion, the authors have provided a graduated color bar of these illustrations. This allows the readers to better visualize the magnitude of the electric field coupling into the waveguides.

7. The authors have provided the spectral of their superluminescent diodes to show the correspondence between the sensitivity of their spectrometer and the illumination provided by their light source. This will clarify the interpretation of their spectra.

In summary, my assessment is that the authors have sufficiently improved their original manuscript according to my criticisms and suggestions. The revised manuscript is more measured, but also more accurate in its claims. The additional experiments and results bolster its scientific credentials. I thank the authors for their careful reading of my review and graceful acceptance of my critiques. With these revisions, I believe the manuscript is scientifically sound and presents novel research of great interest to the wide readership of Nature Communications.

Dear Reviewers,

*Thank you so much for your valuable comments. We feel that the resulting amendments to our manuscript have substantially increased the quality of it. Below, please find our responses to the queries of the reviewer 1 and 2. Note that we use **blue text** for our responses to the reviewers, **red text** when quoting text from the manuscript, and **highlighted red text** to identify new or modified text in the manuscript. We also attach the revised manuscript with highlights that identify the changes.*

Reviewer #1 (Remarks to the Author):

This manuscript presents a fully packaged sensing chip for spectroscopic metrology. The authors point out three requirements for current spectral analysis applications: high resolution, high accuracy and ultra-wide bandwidth. The spectrometer proposed in the manuscript achieves high performance with a cascaded custom MRR, and the results of two applications tested by the authors are also convincing. But I think there are still some issues that need to be addressed before considering publishing this manuscript:

We sincerely appreciate your detailed and constructive comments. We believe the manuscript is significantly improved by the revisions outlined below in response to your comments.

1. The idea of a cascade MRR presented in this manuscript seems to be similar to that of the authors' previous paper [Ref.1], can the authors explain what innovative breakthroughs are made in the design of this manuscript compared to the previous ones.

[Ref.1]: Yao, C. et al. Integrated reconstructive spectrometer with programmable photonic circuits. Nat. Commun. 14, 6376 (2023).

Many thanks to your question. In our 2023 work [Ref.1], programmable unbalanced MZIs were utilized as reconfigurable filters to construct the reconstructive spectrometer. However, the symmetric directional couplers (DCs) with a fixed coupling ratio used in the MZIs posed a tight limitation on its working bandwidth. We made careful examination of that design and achieved a record-high operational bandwidth of 200 nm at that time, but there is little room for further improvement due to the on-chip dispersion.

Thus, in this work, we introduce MRRs where the coupling efficiency of the DC is not stringently maintained. Overall, the waveguide dispersion primarily affects the MRR's round-trip loss and the coupling efficiency of DC. Therefore, to overcome the dispersion limit, we equip the MRRs with curved DCs, which offer greater wavelength stability than conventional symmetric DCs, and then strategically engineer them to operate in an over coupling state. This over coupling state not only generates effective spectral perturbations with minimal loss, but also allows for an ultra-broad bandwidth by balancing the wavelength-dependent loss and coupling efficiency (please find more detailed discussions in our following responses).

To achieve this, we deploy a pair of waveguide bends with a relatively small radius (R_2) on the MRRs to ensure that their bending loss dominates within the ring, such that the round-trip loss would naturally increase at longer wavelength. In accordance, the curved DC is designed to also exhibit an increasing coupling efficiency over wavelength (as shown by Fig. 2(d)), so as to match the rising round-trip loss. As a result, these sophisticated designs jointly enable our spectrometer to realize an ultra-wide bandwidth of over 520 nm.

In addition, by temporarily modulating the MRRs' phase combinations to create a large number of decorrelated sampling channels, we achieve an ultra-high resolution of 8 pm. This corresponds to a bandwidth-to-resolution ratio exceeding 65,000, representing the highest performance reported for on-chip spectrometers. Furthermore, we fully package the photonic chip into an integrated NIR spectroscopic sensor—the first of its kind—and demonstrate a series of NIR applications on various solid and liquid samples, all achieving approximately 100% accuracy. Notably, our sensor realizes a detection limit comparable to that of the commercial benchtop counterparts, establishing a new benchmark in the field.

To better highlight these innovative breakthroughs, we have made revisions to the Introduction, which now reads:

In this paper, we present an ultra-high-performance integrated RS that empowers NIR spectroscopic metrology. Our design simply consists of a cascade of tunable micro-ring resonators (MRRs) equipped with curved directional couplers (DCs). These MRRs are dispersion-engineered to operate under an over coupling state by balancing their wavelength-dependent round-trip loss and coupling efficiency, thereby creating efficient sampling responses across an ultra-wide wavelength range. Meanwhile, the sampling response is temporally decorrelated by manipulating the phase of each MRR. The RS chip is implemented on a SiN integration platform, and gets fully packaged into a chip-scale sensor with auxiliary electronics, demonstrating an over 520 nm operational bandwidth from 1180 nm to 1700 nm, as well as a superior resolution of below 8 pm. This corresponds to a bandwidth-to-resolution ratio exceeding 65,000, which, to the best of our knowledge, is significantly higher than any reported miniaturized spectrometer. A series of NIR spectroscopic applications is demonstrated, including the classification of plastic and coffee samples and the concentration measurement of aqueous and organic solutions, all achieving approximately 100% accuracy. Most importantly, the detection limit of our sensor is examined using glucose solution with concentrations as low as 0.1% (i.e. 100 mg/dL or 5.55 mmol/L) identified. Such level of detectability is already comparable to that of commercial benchtop spectrometers, establishing a new benchmark for NIR spectroscopy with miniaturized sensors.

2. It is mentioned that the MRR is designed to be over-coupled, but the reason is not given, and it is suggested to elaborate in more detail.

We apologize for any confusion in the original manuscript. Following the above discussion, here, we list the detailed reasons for choosing over-coupled MRRs, as:

- 1) Over-coupled MRRs exhibit a larger full-width at half maximum (FWHM) in their resonance peaks compared to critical or under-coupled MRRs. These wider resonance peaks introduce more pronounced spectral perturbations in the overlaid channel responses, thereby improving sampling efficiency.
- 2) Over-coupled MRRs can feature moderate extinction ratios (i.e. a moderate value of T_{res}), ensuring spectral fluctuations with sufficient intensity contrast while introducing minimal excess loss.
- 3) Most importantly, as shown by Fig. 2c, there is a broad design space for parameter combinations (i.e. the loss and coupling coefficient a and r) that allows the MRR to sustain an effective over-coupling state. Hence, by balancing the wavelength-dependent loss and coupling efficiency, the MRR can consistently operate in the targeted over coupling region across a broad wavelength range.

Accordingly, we have made the following revisions for better clarity:

Instead of operating under the stringent critical coupling condition as high-Q narrowband filters, the MRRs in our design are tailored to be over-coupled, targeting to feature resonance peaks with large full widths at half maximum (FWHMs) and moderate extinction ratios (i.e. moderate values of T_{res}). This yields substantial spectral perturbations in the overlaid response to ensure high sampling efficiency, while introducing minimal excess loss. Figure 2c shows the calculated T_{res} under different combinations of α and r . The inset highlights the broad design space where an MRR can sustain an effective over-coupling state, while keeping the T_{res} in a preferable range. Therefore, the key of our MRR design lies in maintaining a balance between the loss and coupling efficiency across a widest possible wavelength range.

3. I don't quite understand the reason why line 103 mentioned that the round-trip loss increases with wavelength.

Again, as discussed above, we strategically employ a pair of waveguide bends with a relatively small radius (R_2) on the ring to ensure that their bending loss dominates the round-trip loss. Since the bending loss increases with wavelength, the round-trip loss of the MRR naturally rises at longer wavelengths. In accordance, we engineer the curved DC to exhibit a gradually increasing coupling efficiency, so as to match the rising round-trip loss, thereby maintaining the MRR in an ideal over-coupling state across an ultra-wide bandwidth.

To avoid any confusion to the readers, we have now revised the relevant text as follows:

For this purpose, we strategically deploy a pair of waveguide bends with a fixed but relatively small radius (R_2) to ensure the bending loss dominate the losses on the ring, so that the round-trip loss naturally increases with longer wavelengths. Meanwhile, we adopt a curved DC structure for greater wavelength stability, thanks to its enhanced phase-matching capability⁴⁴. Its structural parameters are globally optimized using the particle swarm optimization (PSO) algorithm to obtain a gradually increasing cross-coupling efficiency over wavelength, thereby matching the rising on-ring loss.

We have also revised the legend of Figure 2, as:

Figure 2 | Spectrometer design and simulations ... (d) Simulated coupling efficiency of the optimized curved DC, showing an increasing cross-coupling efficiency over wavelength to match the rising round-trip loss on the ring.

4. In Figure 2(e), most of the short-wavelength light remains in the bus, and only a small amount is coupled to the micro-ring, so how to effectively decoupage the light?

As previously elaborated, in our RS, each MRR is intended to introduce distinct yet moderate level of spectral perturbations, i.e. with moderate extinction ratios (unlike those high-Q MRRs in narrowband filtering spectrometers). This, thereby, allows the whole cascaded system to produce overlaid spectral responses with rapid fluctuations and minimal loss for efficient sampling. Accordingly, the coupling efficiency of the DC is designed to match the round-trip loss, ensuring that the MRR sustains in the desired over-coupling state (i.e., with the combination of the coefficients a and r falling within the target region, as shown in Fig. 2c). Therefore, since the round-trip loss at short wavelengths is small, only a small amount of light needs to be coupled. Conversely, at longer wavelengths, where the MRR's loss rises, the

coupling efficiency increases correspondingly. Figures 2d,e vividly illustrate this gradual increase in coupling efficiency over the wavelength.

5. In Figure 4(f), the calculation time of the vertical axis is the result after normalization. Could the authors provide the order of magnitude for the computation time?

We appreciate the valuable suggestion and have updated the vertical axis to reflect the actual computation times. These times are calculated by running the CVX algorithm in MATLAB on a workstation equipped with a Xeon 10980 CPU and 64 GB of memory.

The figures and relevant descriptions have been revised as follows:

Figure 4f plots the reconstruction error and computing time as a function of the channel number (using a Xeon 10980 CPU with 64 GB of memory), and highlights the reconstructed broadband spectra under different channel numbers.

6. Image clarity is relatively low. The color distinctions between different curves in the right-side image of Figure 4(f) are not apparent.

Many thanks for the valuable suggestion. We have redrawn the figure with improved color contrast to enhance the visibility of the curves, as shown below:

7. What is the power consumption of this structure, given that the thermo-optic coefficient of the SiN material is relatively low?

Thanks for the great question. In our 6-stage RS, the phase of each MRR is tuned into four states between 0 and 2π , thereby creating a total of $4^6=4096$ sampling channels. The maximum power consumption of each MRR experimentally measures around 80 mW (i.e., 34 mW/rad). Therefore, the system reaches its maximum power consumption of approximately 480 mW when all MRRs are tuned to the highest phase state. Notably, since such extreme case happens only once over the 4096 sampling channels, the average system power consumption represents better reflection of the overall efficiency, which is half of the maximum, i.e., 240 mW. This power consumption can be further reduced by incorporating deep trenches or undercuts to the waveguide for better thermal efficiency.

We have included relevant discussion in the revised manuscript, as:

A microscope image of the RS chip is shown in Fig. 3c, with the insets magnifying three tunable MRRs with curved DC. The MRRs each occupies a footprint of less than $80 \times 150 \mu\text{m}^2$ and are laterally spaced by 200 μm to minimize thermal crosstalk. We set four phase states per MRR, creating a total of 4096 temporal sampling channels. The power consumption of each MRR experimentally measures around 34 mW/rad, resulting in an average system power consumption of 240 mW. This can be further enhanced by incorporating deep trenches or undercuts in the waveguide to improve thermal efficiency.

8. In line 174, it is mentioned that the plastic particles' spectra have a moderate roll-off ratios, so only 312 sampling channels were used. Does this mean that the higher the roll-off ratio of an unknown spectrum, the faster the corresponding spectral fluctuations, and thus more sampling channels are required?

We appreciate this insightful question. As shown in Fig. 4f-g, we demonstrate that the global sampling feature of RS allows a fine reconstruction accuracy to be maintained when using fewer sampling channels, which, in turn, largely relaxed the computational complexity. This flexibility provides a user-definable performance trade-off to suit different application scenarios. Hence, given that the NIR spectra of typical materials, such as plastics, exhibit gradual changes without rapid roll-offs, we downsize the sampling channel number to 312 is to realize the best balance between the computation time and reconstruction performance, avoiding unnecessary redundancy. In fact, most NIR spectroscopic applications only require sub-nanometer resolution rather than picometer-level (please see Fig. 1b), making 312 channels a suitable choice. Nevertheless, as the reviewer correctly pointed out, for those unknown spectra with narrowband peaks or sharp fluctuations, such as laser signals, it is indeed necessary to increase the channel number to ensure sufficient resolution.

Accordingly, we have made relevant revisions to enhance clarity, which are as follows:

We repeatedly measure each sample for 60 times over the entire 520 nm bandwidth, and randomly split the data into training and testing sets in a 7:3 ratio. Note that for all these measurements, we downsize the sampling channel number to 312 as a balanced choice between computational cost and reconstruction performance (please see Fig. 4f,g), given that the spectra under test exhibit modest roll-off ratios.

9. The determination coefficient of random forest algorithm is suggested to be explained.

We appreciate the suggestion, and have now included the explanation of determination coefficient in the main text, as:

Accordingly, we train a prediction model using the random forest (RF) algorithm (see Methods for modelling details)⁴⁷. Here, the coefficient of determination (R^2) is used to assess model's fitting performance, defined as $R^2 = 1 - \frac{\sum_i (y_i - \hat{y}_i)^2}{\sum_i (y_i - \bar{y})^2}$, where y_i and \hat{y}_i denote the true and predicted concentration value of sample i in the testing set, respectively, and \bar{y} represents the mean of the true value. As a result, our RF model achieves a coefficient of determination (R^2) of 1.000 on the testing sets, indicating an excellent fit.

10. In the method, the authors mention that when designing the DC, the coupling efficiency should gradually increase with wavelength. What is the purpose of this approach?

We apologize once again for the lack of clarity in our original manuscript. As discussed in our responses to comments 1-4, the increasing coupling efficiency in the DC is designed to match the rising round-trip loss in the ring, thereby maintaining the MRR in an over-coupling state across an ultra-broad wavelength range. For more details and relevant revisions, please refer to those responses.

Reviewer #2 (Remarks to the Author):

In the paper, the authors develop a miniature NIR reconstructive spectrometer with 520nm bandwidth and resolution of 8pm. They use this device to demonstrate the identification of several chemicals. Some key innovations are, the use of SiN instead of SIO to reduce material dispersion, moderate extinction ratios for resonators to reduce loss, and using random variation in the free spectral ranges of the resonators to increase randomness to allow decorrelation of different channels. Overall, the resolution, bandwidth, and size of this device make this an important step forward for the community.

We are sincerely grateful for your valuable comments. Accordingly, we have undertaken careful revisions to enhance the quality of our paper. Detailed responses are provided below.

Scientific Inquiries

1. Are the authors confident in their claim of 520 nm bandwidth for their design? The data presented in Figure 3 d shows severe noise at 1670 nm and 1200 nm, this would reduce the operating bandwidth to 470 nm, a roughly 10% decrease from the original claim. Can the authors provide some evidence for the full spectral range of their device and explain how they are able to overcome the high noise at these wavelengths to discern the 8pm resolution?

We sincerely appreciate your insightful question. We are confident in our claim of a bandwidth exceeding 520 nm, as evidenced by both simulation and experimental results. First of all, the core principle behind our design is sustaining an ideal over-coupling state of MRRs across an ultra-wide wavelength range, thereby creating rapid and random spectral fluctuations to enable efficient spectral sampling. This is achieved by dispersion-engineering the curved DC to enable a gradually increasing coupling efficiency over wavelength (see Fig. 2d, e), matching the rising round-trip loss in the ring. To better illustrate this, we've now added the FDTD-simulated spectral response for an optimized MRR to the Fig. 2, showing that it maintains in the desired over-coupling state across an ultra-wide wavelength range from 1160 nm to 1740 nm. Such result suggests the device's bandwidth could actually exceed 580 nm or even more.

While in our experiments, limited by the bandwidths of the four SLD sources and the measurement range of our benchtop spectrometer, we only performed the calibrations and spectral recovery only in a 520 nm range between 1180 nm and 1700 nm. We acknowledge that in the original Fig. 3d, the regions below 1200 nm or above 1670 nm appeared to show noise. This was partly due to the lower optical power of SLD sources in those bands, for which we have utilized the highest sensitivity of our benchtop spectrometer to improve the signal-to-noise ratio. Moreover, the appeared noise can be largely attributed to issues in figure plotting, where insufficient resolution was used during export. We also applied transparency the response curves, which caused significant overlap, making them appear noisy. To address this, we have redrawn Fig. 3d to more clearly present the channel responses over the 520 nm bandwidth.

Accordingly, the relevant revisions are as follow:

Figure 2d,e present the finite-difference time-domain (FDTD) simulated coupling efficiency and light propagation profiles of the tailored curved DC, respectively, showing a rising cross-coupling ratio over wavelength. Besides, each MRR is equipped with a pair of straight waveguides with varying lengths (L_i) to not only achieve small FSRs for rapid spectral roll-offs, but also to break any periodicities in the overlaid response. For example, Fig. 2f plots the

FDTD simulated spectral response for one of the cascaded MRRs, demonstrating that the desired over-coupling state is well maintained over an ultra-broad wavelength range from 1160 nm to 1730 nm.

Figure 2 | Spectrometer design and simulations ... (f) Simulated spectral response for one of the cascaded MRRs, showing the targeted over coupling state across an ultra-wide wavelength range.

Figure 3 | Device images and measured channel spectral responses ... (d) Representative examples of the measured channel spectral responses between 1180 nm and 1700 nm. The inset depicts the emission spectra of the four SLD sources. P.D.: power density. (e) Channel responses at different observation windows.

2. The authors have chosen to use 6 MRRs in their design, can the authors discuss the reason

Many thanks to this valuable question. The choice to cascade 6 MRRs is a balanced choice between the spectrometer performance, system footprint/complexity, and the consumption of sampling channels. Specifically, cascading more MRRs increases the density of spectral perturbations and helps better break the periodicity in the overlaid response, thereby enhancing both the channel decorrelation and sampling efficiency. However, this comes at the cost of increased footprint, power consumption, and control complexity. In our design, 6 MRRs are sufficient to deliver ultra-high resolution down to the picometer scale, along with excellent reconstruction accuracy. On the other hand, implementing a fewer number of MRRs would lower the channel sampling efficiency, such that it would demand a larger number of sampling channels to deliver an equivalent resolution and accuracy. This, in turn, leads to longer sampling times and higher computational complexity. Therefore, the selection of 6 MRRs reflects a careful trade-off between these factors.

To better clarify this, we have made following revisions:

Figure 3a presents our fully packaged NIR spectroscopic sensor at centimeters scale, incorporating the SiN RS chip and a high-speed driving board with a microcontroller unit (MCU) integrated. The insets enlarge two optical sampling interfaces that are tailored for reflective and transmissive measurements, respectively. More details regarding the driving board and sampling interfaces are provided in Methods. **The RS chip is designed to incorporate six cascaded MRRs as a balanced choice between device performance, system footprint/complexity, and the consumption of sampling channels.**

3. There is some evidence of spectral reconstruction provided in Figure 4a and b using lasers as inputs. All the peaks presented here have side lobes or peaks of notable intensity at adjacent wavelengths to the main peaks. Can the authors specify whether these peaks are an artifact of the input light, so an artifact of the device or reconstruction algorithm.

We appreciate your observation. The non-zero lobes/peaks in Figures 4a and b are the results of minor reconstruction errors caused by inevitable measurement inaccuracies. Specifically, the linewidths of the laser signals are well below 0.1 pm (i.e., <10 MHz), far narrower than our spectrometer's resolution. Thus, the reconstructed laser spectra should ideally show energy at only one spectral pixel, with all other pixels being zero. However, given that the reconstruction algorithm processes the whole input spectrum globally, any measurement noise could result in unwanted non-zero pixels across the entire bandwidth, though their amplitudes are typically insignificant compared to the main peak, depending on several factors such as the design of the channel responses, the algorithms used, and the level of measurement error. It should be noted that these non-zero peaks may not necessarily appear near the laser signal but can occur anywhere due to the randomness of the measurement error. This phenomenon is common in computational spectrometers and has been widely documented in the literature (for examples, please see Ref. 3 to 8). In our experiments, all reconstructed laser signals (including those tri-peak signals) exhibit low relative errors ranging from 0.04 to 0.11, demonstrating the superior accuracy of our sensor.

To better clarify this for readers, we have made the following revisions:

As shown in Fig. 4a-c, our RS precisely resolves the intensities and locations for all peaks, exhibiting low relative errors ranging from 0.04 to 0.11. **The minor non-zero peaks apart from the laser signals can be attributed to the reconstruction errors caused by inevitable measurement inaccuracies, which can be further suppressed by enhancing the system's signal-to-noise ratio and applying more advanced reconstruction algorithms.** Notably, the spectral spacing of dual-peak signals is gradually reduced down to 8 pm, marking its resolution according to the Rayleigh criterion.

4. The authors should provide some more details on how the computation cost in their Figures 4f and g are calculated. Is this a computational time metric or number of units required etc.?

We appreciate your valuable suggestion and have now revised Fig. 4f and 4g to reflect the actual computation times (in seconds). These times are calculated by running the CVX algorithm in MATLAB on a workstation equipped with a Xeon 10980 CPU and 64 GB of memory.

The figures and relevant descriptions have been revised as follows:

Figure 4f plots the reconstruction error and computing time as a function of the channel number (using a Xeon 10980 CPU with 64 GB of memory), and highlights the reconstructed broadband spectra under different channel numbers.

5. While the spectral reconstruction is quite accurate, there is a consistent underestimation of the signal at ~1530-1535 nm for Figures 4d and e. The consistency of this error is puzzling. Can the authors comment on whether this is a system artifact?

Thank you for the inquiry. In theory, given that RSs employ a global sampling strategy to recover the incident spectrum by solving underdetermined equations, any measurement errors will cause reconstruction errors distributed across the entire reconstructed spectrum rather than affecting specific regions. Meanwhile, due to the randomness of the measurement errors themselves, the resulting reconstruction errors are also random, leading to either underestimations or overestimations. For example, in the 1530-1535 nm range as mentioned by the reviewer, it can be noted that the reconstructed spectral peak shows an underestimation in the first two plots of Figure 4(d), but an overestimation in the remaining two plots. This indicates that the variations are simply reconstruction errors due to measurement inaccuracies, rather than a system artifact or design flaw.

6. In Figure 5, the reconstructed spectra are provided along with the reference spectra of various samples. While the spectra match reasonably well, it would be useful to have a quantitative error measurement with each of the presented samples. Also, are the authors able to provide the maximum allowable error for their algorithm to predict the sample correctly? These two metrics will help distinguish the fidelity of the device from the predicting power of the algorithm.

We sincerely appreciate these constructive suggestions. In response, we have calculated the average relative error and standard deviation from the 60 tests on each plastic and coffee sample, and summarized the results into an additional table S1 in Supplementary Material Section 4. The results show that the measured plastic and coffee samples exhibit average relative errors ranging from 0.026 to 0.054 and 0.031 to 0.056, respectively. Additionally, the calculated standard deviations for all samples remain around 0.02. These findings demonstrate the exceptional precision and stability of our sensor.

To test the model's maximum allowable error, we generate a series of spectra with varying levels of relative error by artificially superimposing random, continuous spectral errors of different magnitudes to the reference spectra of each sample. Following the same model training process, we then obtain the model's accuracy under different levels of spectral error, as shown in the additional Fig. S5. The results indicate that for plastic samples, the model accuracy begins to deviate from 100% when the average relative error reaches approximately 0.08, and it drops to 70% when the relative error increases to around 0.64. Similarly, for coffee samples, model accuracy cannot be maintained at 100% once the relative error exceeds about 0.06, and it drops to about 60% when the relative error surpasses around 0.55.

Accordingly, we have made the relevant revisions in both the main text and Supplementary Material Section 4, as follows:

Main text:

Figure 5a,b show the measured reflectance of two representative samples, respectively, for the plastic and coffee. The reflectance spectra for all remaining samples are provided in Supplementary Fig. S4. The average spectral relative errors for these plastic and coffee samples range between 0.026 to 0.056, as shown by Supplementary Table S1. The minor discontinuities in the measured spectra are due to the low SLD power density at those wavelength regions (see Fig. 3d). Corresponding classification models are trained on the basis of support vector machine (SVM)⁴⁵ to identify plastic or coffee samples. Figure 5c,d present the classification results for the plastic and coffee samples, respectively, all demonstrating 100% accuracy. A detailed investigation into the relationship between model prediction accuracy and reconstruction errors is also conducted via simulations, as shown by Supplementary Fig. S5. The analysis reveals that the 100% accuracy begins to decline when relative errors exceed certain thresholds, with turning points for plastic and coffee samples occurring at around 0.08 and 0.06, respectively.

Supplementary Material Section 4:

Figure S4a,b present the reflectance spectra of the remaining eight plastic and coffee samples, respectively, showing that our sensor maintains excellent consistency during repeated measurements. In Fig. S4c,d, we summarize the reflectance spectra for all ten types of plastic and coffee samples within specific observation windows, respectively. Subtle differences in NIR spectral features can be observed among different samples. To better quantify the reconstruction accuracy, Table S1 outlines the average relative error and standard deviation from 60 tests on each plastic and coffee sample. As shown, the measured plastic and coffee samples exhibit average relative errors ranging from 0.026 to 0.054 and 0.031 to 0.056, respectively. Additionally, the calculated standard deviations for all samples remain around 0.02. These results demonstrate the excellent precision and stability of our sensor.

To investigate the relationship between model accuracy and spectral reconstruction error, we generate a series of spectra with varying levels of relative error by artificially superimposing random, continuous spectral errors of different magnitudes onto the reference spectra of each sample. Following the same training process, we then evaluate the model's prediction accuracy under different levels of relative error, as shown in Fig. S5. The results indicate that for plastic samples, model accuracy begins to drop from 100% when the average relative error reaches approximately 0.08, decreasing to 70% at a relative error of around 0.64. Similarly, for coffee samples, accuracy can no longer be maintained at 100% once the relative error exceeds about 0.06, dropping to around 60% when the relative error surpasses 0.55.

Figure S5. Relationship between the model prediction accuracy and spectral relative error for plastic and coffee samples.

Table S1. Average relative error and standard deviation of tests for plastic and coffee samples

Plastic sample	Mean	Standard deviation	Coffee sample	Mean	Standard deviation
ABS	0.0323	0.0236	CMT	0.0414	0.0206
AS	0.0537	0.0266	DZW	0.0376	0.0247
PBAT	0.0463	0.0260	HFH	0.0429	0.0228
PET	0.0378	0.0253	HK	0.0317	0.0269
PHB	0.0457	0.0214	HYY	0.0488	0.0215
PP	0.0261	0.0182	KX	0.0305	0.0268
PTEE	0.0530	0.0314	XDM	0.0473	0.0291
PVC	0.0540	0.0241	YNPE	0.0524	0.0257
PC	0.0373	0.0154	HHZ	0.0455	0.0204
PVA	0.0424	0.0168	LWD	0.0560	0.0195

7. While the detection limit and bandwidth of the presented device are comparable to commercial spectrometers according to Table 1, the mean absolute error and root mean squared error are substantially higher. Could the authors comment on the reason for this and the following? How is the spectrometer able to achieve similar detection limits while having higher errors? What is the limitation caused by this error?

We appreciate the insightful questions. The higher mean absolute error (MAE) and root mean squared error (RMSE) in our concentration tests compared to the results obtained via commercial benchtop spectrometers can be attributed to the differences in measurement stability/repeatability. Specifically, our chip-based sensing system exhibits a relatively larger variance in measurements, meaning that while the majority of tests achieve high accuracy, a few individual tests may show higher errors. Meanwhile, since we test a wide range of concentrations from 0.1% to 40%, and both MAE and RMSE are absolute error metrics, they naturally place more weight on those measurements with higher concentrations. For example, a 1% error when measuring a 10% concentration solution contributes significantly more to the overall MAE and RMSE than a 0.01% error when measuring a 0.1% concentration solution, even though their error proportions are the same (i.e. $\frac{1\%}{10\%} = \frac{0.01\%}{0.1\%}$). Therefore, while the detection limit remains comparable, as it reflects the system's ability to detect the smallest

concentration, the relatively larger variance in our sensor's measurements—particularly in the high concentration tests—results in higher MAE and RMSE values.

It is also important to note that the difference in measurement stability is not solely due to the spectrometers themselves, but also because of the different light sources used in the experiments. In fact, the concentration measurements with the benchtop spectrometers (both Bruker MPA II and IdeaOptics NIR17+Px) were conducted using commercial broadband NIR sources (based on tungsten-halogen lamps), which are more stable than the SLD sources, leading to lower MAE and RMSE values. Therefore, the measurement stability of our sensor can be improved by optimizing the chip control system for better precision and repeatability, as well as by employing more stable light sources.

Accordingly, we have added the relevant elaborations and discussions in the revised manuscript, which read as follows:

We replicate the glucose concentration tests using benchtop dispersive and FT spectrometers (IdeaOptics NIR17+Px and Bruker MPA II), in tandem with commercial broadband NIR sources based on tungsten-halogen lamps. Note that here we construct different prediction models using their full or partial operational bandwidth for fair comparisons (see Supplementary Fig. S5 for more details). Table 1 details that our NIR sensor reaches the detection limit and accuracy comparable to those of commercial counterparts. The relatively higher MAE and RMSE can be attributed to the difference in measurement stability, which can be improved by optimizing the chip control system and employing more stable light sources.

Presentation Notes

8. In Figure 2e, the authors illustrate the simulated light paths through their device. Can the authors please add an annotation of the color bar to indicate the units depicted and whether this is simply a normalized scale with arbitrary units?

Thanks for pointing this out. The color bar represents the normalized electric field intensity. Accordingly, we have revised Fig. 2e to include the annotation, and updated the figure legend, as:

Figure 2 | Spectrometer design and simulations ... (e) Simulated light propagation profiles in the curved DC at different wavelengths, shown as normalized electric field intensities.

9. In Figure 5, it would be useful to provide the spectrum of the source in an inset to prevent confusion about the missing spectral bands Figure 5b. If possible, the authors are requested to provide some measurements with a broadband source to avoid this issue.

We thank for the constructive suggestions. In our original manuscript, the measured spectra of all SLD sources are actually provided in Supplementary Fig. S7b. Yet, we fully agree that including them in the main text would better prevent confusion for readers. Thus, we have now

modified Fig. 3c to include these source spectra. On the other hand, since most commercially available NIR broadband sources are centred around 1310 nm and 1550 nm, it is, unfortunately, both technically challenging and expensive to customize additional light sources to fulfil these missing bands.

The corresponding revisions are as follow:

For the calibration of RS chip, four superluminescent diodes (SLDs) centered at different wavelengths are introduced as light sources (see more discussions in Methods). Figure 3d plots representative measured sampling channels over a 520 nm bandwidth between 1180 nm to 1700 nm (Supplementary Fig. S1 shows the whole sampling matrix). The inset shows the emission spectra of the four SLDs. Fig. 3e further display the channel responses at four different observation windows, highlighting the random spectral fluctuations.

Figure 3 | Device images and measured channel spectral responses ... (d) Representative examples of the measured channel spectral responses between 1180 nm and 1700 nm. The inset depicts the emission spectra of the four SLD sources. P.D.: power density. (e) Channel responses at different observation windows.